# Geographical–Historical Analysis of the Herbarium Specimens Representing the Economically Important Family *Amaranthaceae* (*Chenopodiaceae-Amaranthaceae* Clade) Collected in 1821–2022 and Preserved in the Herbarium of the Jagiellonian University in Krakow

**DOI:** 10.3390/biology13060435

**Published:** 2024-06-13

**Authors:** Agata Stadnicka-Futoma, Marcin Nobis

**Affiliations:** 1Department of Soil Science, Environmental Chemistry and Hydrology, Institute of Agricultural Sciences, Environment Management and Protection, College of Natural Sciences, University of Rzeszów, ul. Zelwerowicza 8b, 35-601 Rzeszów, Poland; 2Institute of Botany, Faculty of Biology, Jagiellonian University, Gronostajowa 3, 30-387 Kraków, Poland; m.nobis@uj.edu.pl

**Keywords:** *Amaranthaceae*, *Chenopodiaceae*, herbarium collection, KRA, vascular plant, taxonomy, biodiversity

## Abstract

**Simple Summary:**

The digitization of herbarium collections is an important process that allows access to sometimes huge amounts of data that can be used in various natural sciences. In recent years, the collections of one of the thirty oldest herbariums in the world—the herbarium of the Jagiellonian University—have been digitized. This paper presents the resources of the economically very important *Amaranthaceae* family, which include 8801 herbarium sheets. They were analyzed in taxonomic, geographical, historical, and functional terms.

**Abstract:**

Herbaria constitute a form of documentation, store and secure comparative material, as well as constitute an extra original gene bank. They are an invaluable database among others for the biological, ethnobotanical and agricultural sciences. The digitization of herbarium collections significantly facilitates access to archival materials; however, searching them is still time-consuming. Therefore, our work aims to analyze the herbarium collection of 8801 sheets for specimens representing the economically important family *Amaranthaceae* (*Chenopodiaceae-Amaranthaceae* clade) deposited the oldest herbarium in Poland, the herbarium of the Jagiellonian University (KRA). These specimens have been collected from almost all the continents in dozens of countries for over 200 years. The analyses conducted, including the taxonomic coverage, geographical characteristics and origin, temporal coverage and utility importance of representative species, present the discussed resources in a more accessible way and may become a more attractive form for scientists potentially interested in more advanced research work.

## 1. Introduction

Herbaria, have become increasingly popular. They constitute a form of documentation, store and secure comparative material, as well as constitute an extra original gene bank. Currently, dried plant specimens are commonly used in taxonomic, phytogeographic and cytological research. On their basis, morphometric tests can be performed, which can later be used for statistical analysis (so-called numerical taxonomy) or to describe the morphology and anatomy of individual structures at the micro level. Taxonomists are often able to use digitized collections to identify and annotate specimens [1], to indicate typical specimens and to typify names of species for which the type has not been designated or is regarded as lost or destroyed. In recent years, publications including typification made based on the available online herbarium databases are becoming more and more common [2,3,4,5,6,7,8,9,10]. It is worth noting that herbaria are a source of new species, collected and often set aside as unidentified species. Following Bebber et al. [11], about 70,000 species are still waiting to be described. In the era of digitization, the use of scans or photographs of herbarium specimens greatly facilitates the work as well as significantly saving time [9,12,13,14]. An important aspect is also the studies of phylogenetic relationships. Increasingly, phylogenetic analyses are based on samples collected from herbarium materials, e.g., [15,16,17,18,19,20,21,22], despite problems related to post-mortem DNA damage [23,24]. Herbaria also play an important role in research related to the distribution of plants. At the same time, they allow for analyzing changes in plant cover over many years and are used to establish phytogeographic patterns. Herbarium specimens can be used to analyze phytogeographic phenomena with temporal and spatial dynamics. On their basis, for example, the migration of invasive plants is studied, e.g., [25,26,27,28,29,30,31]. They have similar importance in the case of rare plants, including endangered ones, but also utility plants and weeds. They therefore constitute a biodiversity database and may be indirectly helpful in its protection. Herbarium specimens allow the identification of priority places for nature conservation, e.g., [32,33]. As it turns out, herbaria can be a source of viable diasporas. In recent years, attempts have been made to use the seeds of herbarium specimens, and the success of these studies is essential in the case of endangered or even extinct species [34]. Preserved tissues can also be used for this purpose [35,36]. Based on the data contained on the labels of herbarium specimens, it is possible to reconstruct the phenology of plant species in large areas. Phenological data are, therefore, an important tool for studying the impact of climate change on living organisms [37], particularly those related to global warming [38,39,40,41,42]. Based on the collected specimens or collections, it is also possible to trace the history of botanists, e.g., in relation to the expeditions they undertook, which were previously unknown [36,43]. They are also a source of information for ethnobotanists, e.g., [44].

The herbarium of the Jagiellonian University is one of the thirty oldest herbaria in the world [43]. Currently, it contains collections of vascular plants (approx. 800,000 sheets), bryophytes (25,000 specimens), fungi and slime molds (over 70,000 specimens), lichens (approx. 100,000) and algae (approx. 5000). In addition, collections of cones and fruits numbering over 1000 specimens are deposited there. In total, there are approximately one million specimens from all over the world. In 2019, the digitization of the herbarium’s sheets began as part of the IMBIO project [45,46]. During this time, several large and numerous families of vascular plants were digitized, including one of the most useful and economically important families, *Amaranthaceae.*

Following the APG treatment, the family *Amaranthaceae* is represented by ca. 165 genera, comprising over 2050 species [47]. However, particular members of the family are taxonomically difficult and require integrative studies. It is worth admitting that only in the last decade have a dozen species within the family been described, e.g., [48,49,50,51,52,53,54].

*Amaranthaceae* contains many economically important crop plants, adapted to a wide range of climatic conditions due to the type of photosynthesis carried out, C4 types, but also due to their high nutrient content, e.g., [55,56,57,58,59]. For example, within the genus *Chenopodium*, there are many species, for instance, *Ch. quinoa* Willd., that contains proteins composed of amino acids that are usually missing from cereal grains (e.g., large amounts of lysine), which are considered to be richer in nutrients than cereals [60,61,62]. The genus *Amaranthus* is also known for containing valuable nutrients, a high protein and mineral content even compared to rice or corn grains, and an appropriate amount of exogenous amino acids (including, like the above-mentioned quinoa, lysine) [63]. Most species are also used in medicine. They have numerous medicinal properties, including anti-inflammatory, antirheumatic, diuretic, and in the case of some of them, e.g., *Achyranthes aspera* L., anticancer activity has been demonstrated, e.g., [64,65,66,67,68]. Many studies indicate that, for example, *Atriplex lindleyi* Moq. may contribute to the fight against malaria and be useful in the prevention of *Candida albicans* (C.-P. Robin) Berkhout and *Pseudomonas aeruginosa* (Schröter) Migula [69]. Many species are also used in animal breeding, e.g., species of the genus *Amaranthus* are used in the production of animal feed [70]. Species used in the process of recultivation of contaminated areas are also of particular importance, e.g., *Amaranthus dubius* Mart. ex Thell. (phytoremediation of solid municipal waste landfills) [71,72,73], whose properties can be used to treat soil and increase the crop area. The interest in the agricultural or pharmaceutical use of this group of plants certainly results from the fact that they are also often used for food, medical, cosmetic, textile, construction, religious and even magical purposes in various parts of the world. They are therefore a rich source of research for ethnobotanists [74,75,76,77,78]. Because the family *Amaranthaceae* is such an economically important group of flowering plants, in this work, we decided to analyze the species deposited in the KRA herbarium in order to make the digitized collection available to a wide audience. The main aim of our work was thus the analysis of all of the herbarium sheets with representatives of the above mentioned family, in terms of the taxonomy, the origin of the specimens and the time of their acquisition, which will significantly facilitate more advanced research work. In order to highlight the importance of the *Amaranthaceae* family, the literature has been reviewed. Endangered and rare species, as well as those of economic importance, were indicated.

## 2. Materials and Methods

The subjects of this article are specimens of species from the *Amaranthaceae* family (*Chenopodiaceae-Amaranthaceae* clade) deposited in the KRA Herbarium. During digitization, the information on the herbarium labels was recorded in the appropriate columns of the Jagiellonian University’s MUZUJ database. Moreover, especially in the case of the genus *Chenopodium*, the authors verified the correctness of the identification. A complete list of the specimens studied is presented in Appendix A.

The families currently included in the *Amaranthaceae*-*Chenopodiaceae* clade have long been considered closely related, e.g., [79]. Kadereit et al. [80] listed several common features in terms of the morphology and anatomy, confirming the close relationship of species representing both families. Currently, representatives of both families are grouped into the family *Amaranthaceae*, treating *Chenopodiaceae* as its synonym [81,82,83]. Despite this, both Kadereit et al. [80], as well as many other researchers, e.g., [84,85,86,87,88], suggest that the concept of combining both families has no solid phylogenetic basis and only reflects the fact that they form a monophyletic group [80,84,86,89,90,91]. Therefore, they treat the families in question as the *Chenopodiaceae-Amaranthaceae* clade. Our work is based on a systematic division according to the applicable APG IV [83], while the nomenclature follows the Word Flora Online [92]. If names were not found in the mentioned list, the Plants of the World Online [93] platform was used. Information regarding the life forms of the analyzed plants or from publications cited in the text was also obtained from both sites.

## 3. Results

### 3.1. General Information

The total number of sheets with specimens representing the *Amaranthaceae* family is 8801 (Appendix A). Almost 97% of them are treated in the rank of species and 6% as subspecies or variety. Slightly over 2% are identified as genera, and 0.15% are putative hybrids. The remaining 0.5% constitute specimens with illegible names, belong to the unresolved name category or are assigned to only the family. The geographical location is specified for approximately 92% of the specimens. 81% of the labels contain information on the habitat, while the remainder have no such information or the text is illegible. Generally, only a small part of the sheets provide data such as the geographical coordinates, altitude, inclination, exposure or ATPOL square number (in the case of specimens collected in Poland) [94]. Moreover, 92% of the sheets have the collector’s name and/or surname legible, while the remaining personal details are illegible or non-existent. The collection date is specified for 94% of the specimens, with full dates appearing on 86% of the specimens. In other cases, they consist of only the year, the year and the month, or are illegible. The rest are sheets without a collection date assigned.

### 3.2. Taxonomic Coverage

The specimens stored in the herbarium of the Jagiellonian University and representing the analyzed family belong to 346 species and constitute approximately 17% of the world’s resources [93]. The species with the most significant number of sheets (11% of resources) is *Chenopodium album* L., while 131 species are represented by single specimens.

Generally, species representing the *Amaranthaceae-Chenopodiaceae* clade belong to 83 genera (Figure 1), constituting 47% of the world’s resources [93]. The largest number of deposited specimens represent *Chenopodium* (3385 sheets), *Atriplex* (1905 sheets), and *Amaranthus* (1334 sheets). The remaining genera are represented by less than 100 sheets, and 53 genera are represented by only 1 sheet. *Atriplex*, *Salsola*, *Chenopodium* and *Amaranthus* are represented by the greatest numbers of species, while 41 genera are represented by only 1 species.

### 3.3. Life Forms

Most specimens representing the *Amaranthaceae* family constitute herbaceous plants (77% of annuals and 23% of perennials). Shrubs constitute 22%.

### 3.4. Geographical Characteristics and Origin of Specimens

The specimens representing the *Amaranthaceae* family deposited in the KRA herbarium come from almost all over the world (Figure 2).

The country where the individual plant was collected is specified on 8069 labels. The vast majority of specimens were collected in Europe (88.11%). The percentage of plant origin in relation to individual continents is presented in Figure 3.

Within Europe, the vast majority of specimens (almost 6000) have been collected in Poland, 306 in Ukraine, and 127 sheets in Spain, mostly from the Canary Islands. The Asian specimens representing the family *Amaranthaceae* mainly come from Tajikistan (103 sheets), including the only specimen of *Corispermum papillosum* (Kuntze) Iljin—a species with a relatively narrow range; 72 specimens come from Russia, with almost 67% collected between 1843 and 1933, while a single specimen of *Atriplex gmelinii* C. A. Mey. ex Bong. was collected in South Korea. The African *Amarantahceae* flora are represented by 143 specimens. The greatest number (47) was collected from Kenya (with 46 specimens collected by A. Starzeński in the years 1943–1956). A total of 180 specimens come from North America; however, most of them (135 sheets) were collected in the United States. Single specimens have been collected in Honduras and Jamaica, including the sole specimen of *Froelichia interrupta* (L.) Moq. South America is represented by 23 specimens, collected mainly in Uruguay in the years 1899–1986. A significant part of this collection represents the genera of *Gomphrena*, *Alternanthera* and *Celosia*. A single sheet with specimens of *Iresine angustifolia* Euphrasén comes from Argentina, and four specimens of other species (including the only specimens from *Nitrophila occidentalis* (Moq.) S. Watson, *Atriplex imbricata* (Moq.) D. Dietr., *Atriplex atacamensis* Phil. come from Chile. Two sheets of *Suaeda altissima* (L.) Pall. were collected in Peru by A. Junge from the same place (one in 1907 and the other in 1909). 

An important part of the collection comprises rare and endemic plant species with a limited distribution range. In the KRA herbarium, they are represented by 13 species of which the majority occur in Australia (Appendix A).

### 3.5. Temporal Coverage

Specimens of vascular plants have been collected and preserved in the KRA herbarium since approximately 1750. However, the oldest collection representing the family *Amaranthaceae* (complete gathering and properly labeled), *Gomphrena globosa* L., came from 1843 and is very well-preserved (Figure 4).

Generally, the collection can be divided into three historical periods. In the first period, which took place between 1821 and 1899, 281 specimens were collected, with the largest number of specimens (38) collected in 1876. The largest number of sheets with specimens of *Amaranthaceae* was collected and deposited in KRA by A. Śleńdziński (56 sheets). They come from various regions of Ukraine. Most of the plants collected by Sleńdziński can be classified as rather common species, mostly representing the genera of *Atriplex*, *Chenopodium* and *Polycnemum*. The most frequently collected species were *Atriplex rosea* L. (23 sheets) and *Polycnemum arvense* L. (22 sheets). A total of 4675 specimens come from the period between 1900 and 1999. The greatest number of sheets was collected by H. Trzcińska-Tacik, 689 specimens, most of them from Poland. A small part represents Czech, Slovak and Hungarian flora. A significant number was also provided by P. Szotkowski (233) from Upper Silesia, D. Fijałkowski (220) from the Lublin region, A. Śleńdziński (185) from Ukraine, A. Sendek (142) from Silesia, J. Kornaś (120) from different countries but mostly from Poland and Zambia, K. Towpasz (106) from south-eastern Poland (mainly from the foothills) and M. Wajda (106) mainly from southern Poland. The largest number of sheets contains specimens of the following species: *Chenopodium album* L. (462), *Atriplex patula* L. (405), *Atriplex prostrata* Boucher ex DC. (274), *Chenopodium polyspermum* L. (264) and *Amaranthus retroflexus* L. (257). Moreover, 89 individual species sheets and a specimen of the genus *Cyphocarpa* come from this period. However, 3155 specimens were collected in the last and shortest period, between 2000 and 2022 (Figure 5). Most of them were collected in 2003 (699 sheets) and in 2002 (401 sheets). Over 200 sheets were collected in 2004 and each year between 2006 and 2009. The most significant numbers of specimens were collected by the following botanists: M. Nobis (373 sheets) from Poland, Tajikistan, Russia and Kyrgyzstan, as well as by M. Zarzyka-Ryszka (368 sheets), R. Piwowarczyk (252 sheets), A. Nobis (240 sheets), A. Trojecka-Brzezińska (188 sheets) and M. Bielecki (153 sheets) from Poland but from different regions. In general, 8249 labels share the collector’s name for a given plant. On 144 labels, the names are partially or completely illegible, while on the remaining labels, the names of the collectors are missing. This extensive collection of *Amaranthaceae* specimens is the result of almost 1000 researchers, who within over 200 years collected the plants from different regions of the world. The significant increase in the number of sheets collected after 1950 is related to the more intensive scientific activity of mainly employees (research trips) but also doctoral students (floristic studies) associated with the Jagiellonian University.

### 3.6. Utility Importance of Species Representing the Amaranthaceae Family

Of the 346 species representing the *Amaranthaceae* family in the KRA herbarium, at least 65 have been shown to be of significant economic importance. Almost 74% have healing properties for various diseases, and some even have anti-cancer properties. A total of 25 species belonging to the genera *Atriplex* (14) and *Amaranthus* (10) are used as food. In the case of the latter, the seeds are particularly important as they are treated as “pseudocereals” rich in valuable nutrients [63]. These plants also provide food for animals. Some, e.g., *Aerva javanica* (Burm.f.) Schult. or *Arthrocnemum macrostachyum* (Moric.) K.Koch can be used in phytoremediation and soil reclamation [95,96]. *Alternanthera caracasana* Knuth is a potential source of bioethanol for fuel production [97]. Some species, such as the representatives of the genus *Dysphania* R. Br., are used as moth repellent and treatment for respiratory illness, and as apotropaic and insect repellents, which are blessed during Catholic church holidays [98,99]. The list of useful species representing the family *Amaranthaceae* that are preserved in the herbarium of KRA is provided in Appendix A. 

## 4. Discussion

Plants play a significant role in human life. They have also been an object of interest in many fields of science, and fresh or dry parts of plants are a source of valuable and reliable data. Herbaria preserving a collection of dry plants constitute a rich resource of biological diversity. In addition to their museum, historical and scientific value, they protect comparative material and constitute a specific gene bank whose importance, together with the development of molecular technologies, is increasingly appreciated [36,100]. It seems that the most important task of herbaria is to store herbarium specimens as comparative materials for taxonomic research or studies related to the analysis of plant distribution. However, the role of herbaria is much more complex, and those mentioned above are only a few examples.

Digital collections have become an urgent necessity in today’s large-scale biodiversity research, and indeed, in recent years, an intensive digitization process has been taking place in many herbaria [101]. This makes the collections available to a wide range of recipients. Easily accessible and searchable data on herbarium specimens on the Internet have greatly increased the efficiency of research in the last decade. Time previously spent travelling to collections or waiting to borrow specimens can instead be spent on data collection [9].

Analyses such as ours, as included in this article, contribute to a more accessible presentation of herbarium resources of species representing the *Amaranthaceae* family, which is of great economic importance. We believe that this way of presenting them can encourage researchers to use data from the KRA herbarium for various types of scientific research, such as ethnobotanical, pharmacological, phytogeographical or taxonomical research, for instance, related to one of the species-rich, morphologically variable and taxonomically problematic genera such as *Chenopodium*, which we intend to address in our further research. An accessible analysis, combined with the attached list of deposited perches and online access to the herbarium sheets, which are or will soon be published in the GIBF [102], provides new research opportunities and allows researchers to save time previously spent on the revision of herbarium materials directly in the place where they are stored. 

## 5. Conclusions

In this work, we described the whole set of 8801 herbarium specimens representing the economically important family *Amaranthaceae*, which were collected over 200 years and are preserved in the herbarium of the Jagiellonian University (KRA). The analyses presented here, which are related to the taxonomic coverage, geographical characteristics and origin, temporal coverage, or utility importance of species, allow for a better understanding and contribute to faster familiarization with a huge body of material, the verification of which in its raw form would require a tremendous amount of work and time. In the era of increasing possibilities of using herbarium plants, the collected resources of specimens from the discussed family may become an important source for scientific studies involving many fields of natural sciences. It is worth emphasizing that individual representatives of the family *Amaranthaceae* are regarded as taxonomically problematic and can cause many problems in terms of proper identification. Therefore, the data presented in both descriptive and tabular form will be useful for comparative analyses by professional scientists conducting more advanced research works as well as by students and amateur botanists dealing with plants of the family *Amaranthaceae*.

## Figures and Tables

**Figure 1 biology-13-00435-f001:**
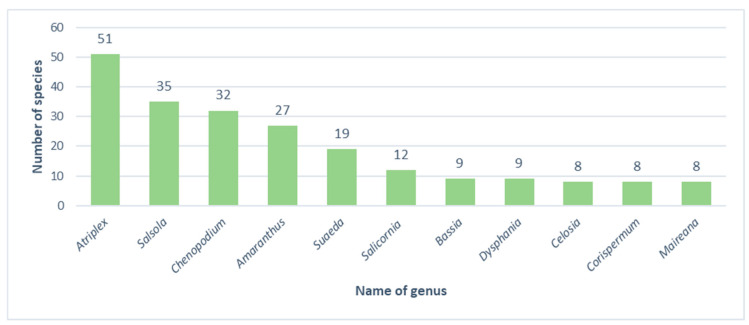
Number of species in the richest genera of the family *Amaranthaceae* in the KRA herbarium (own work).

**Figure 2 biology-13-00435-f002:**
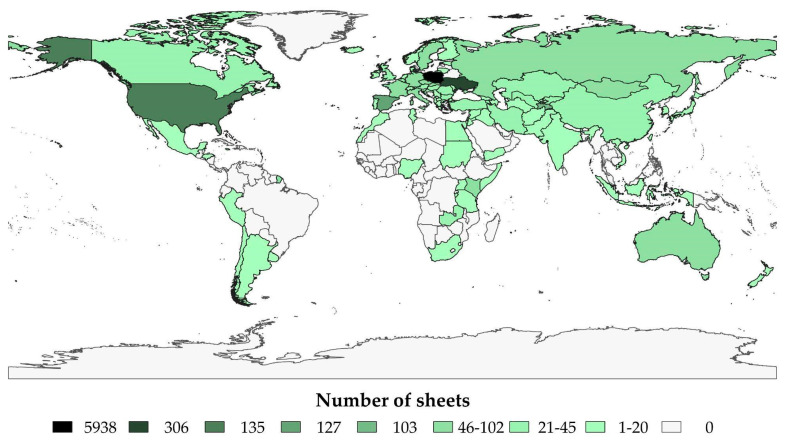
The origin of the specimens preserved in the KRA herbarium (own work based on the WMS EnviroSolution—https://www.envirosolutions.pl/ (accessed on 20 January 2024)).

**Figure 3 biology-13-00435-f003:**
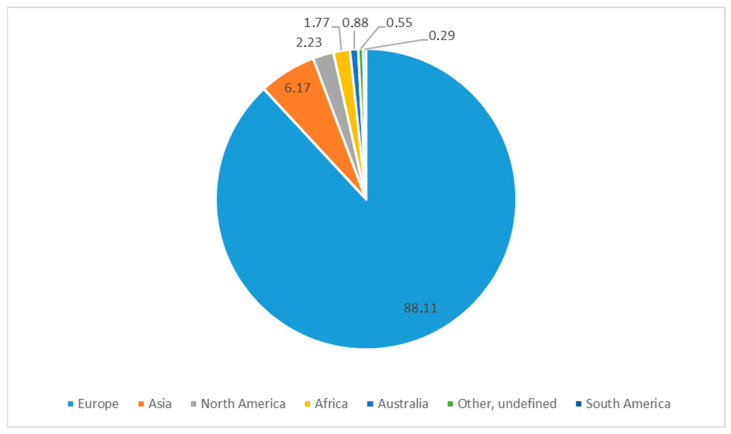
Percentage share of specimens representing the family *Amaranthaceae* collected on different continents and preserved in KRA (own work).

**Figure 4 biology-13-00435-f004:**
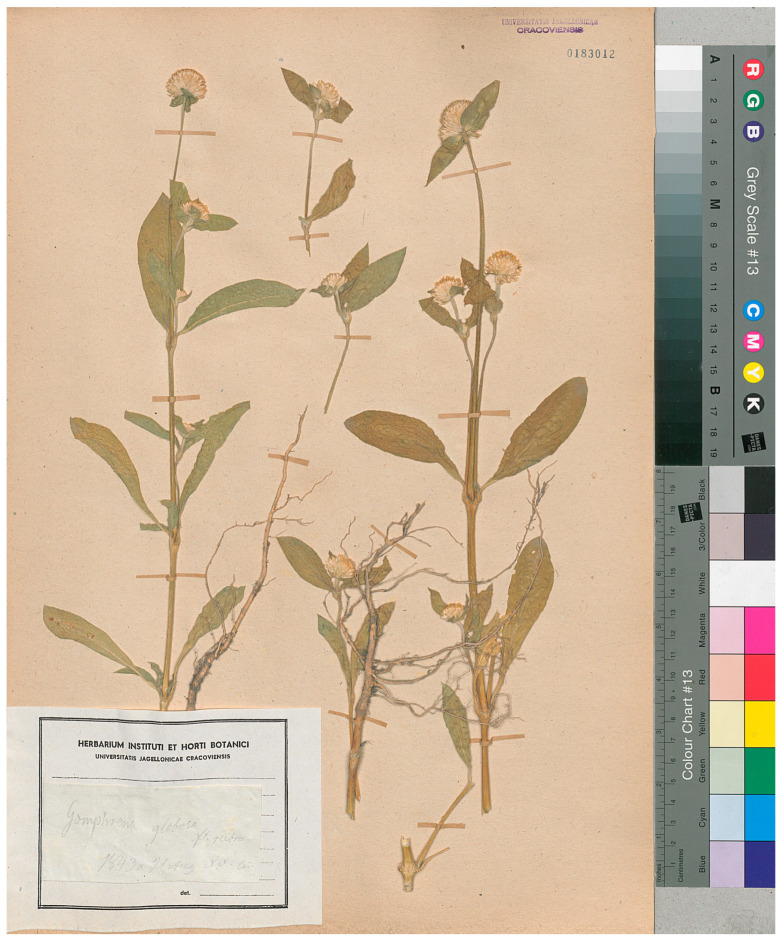
*Gomphrena globosa* L., one of the oldest specimens of the *Amaranthaceae* family preserved in the KRA herbarium (scan of the sheet number UJ-KRA-P-36872-N owned by the herbarium KRA).

**Figure 5 biology-13-00435-f005:**
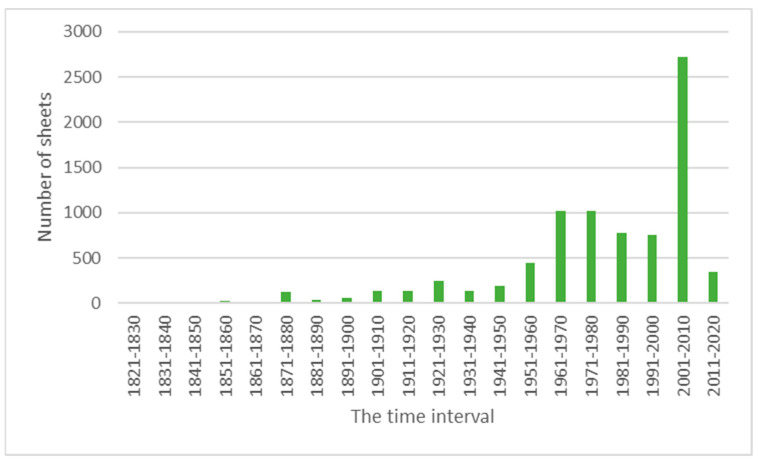
Number of sheets with specimens from the family *Amaranthaceae* collected in individual decades (1821–2022) and preserved in KRA (own work).

## Data Availability

The data presented in this study are available in the Appendix A.

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
