# Peer review of "Geographical–Historical Analysis of the Herbarium Specimens Representing the Economically Important Family Amaranthaceae (Chenopodiaceae-Amaranthaceae Clade) Collected in 1821–2022 and Preserved in the Herbarium of the Jagiellonian University in Krakow"

_biology, 2024, doi:10.3390/biology13060435_

Round 1

Reviewer 1 Report

Comments and Suggestions for Authors

The author of the scientific name of species and genera should be indicated the first time the name is written.

Comments on the Quality of English Language

it is correct, but punctuation marks should be checked.

Reviewer 2 Report

Comments and Suggestions for Authors

Followings are the comments for the improvement of the paper titled “Geographical-historical analysis of the herbarium specimens 2 representing the economically important family Amaranthaceae 3 (Chenopodiaceae-Amaranthaceae clade) collected in years 1821-4 2022 and preserved in the Herbarium of the Jagiellonian Univer-5 sity in Krakow (KRA)”.

The abstract begins with a comprehensive introduction to herbaria and their significance, but it could be more focused on the specific research at hand.

Consider streamlining the introductory sentences to provide a clearer focus on the analysis of the herbarium collection of Amaranthaceae specimens.

Specify the objectives and methodologies employed in the analysis of the herbarium collection. What specific aspects of the specimens are being analyzed (e.g., distribution, morphology, genetic diversity)?

Provide more detailed information on the geographic and temporal scope of the collection. For example, which continents and countries are represented in the specimens, and are there any patterns or trends observed over the 200-year period?

Emphasize the significance of studying the economically important family Amaranthaceae. How does this research contribute to our understanding of agriculture, ethnobotany, or biodiversity conservation?

Highlight the potential practical applications of the research findings. For example, how might this analysis inform agricultural practices or conservation efforts related to economically valuable plant species?

Ensure consistency in terminology and spelling throughout the abstract (e.g., "useful" vs. "usefull").

Improve Figure 2 for readers. Some data of map is not clear.

Re-check the data in Figure three regarding

Consider revising sentence structures for clarity and conciseness. For instance, the sentence "Having in mind the increasing possibilities of use the herbarium materials, documentation of the collection’s composition and origin of the species stored there is now a priority, specially for useful plants and economically important" could be rewritten for better readability and flow.

If applicable, include references to relevant literature or previous studies that support the importance of analyzing herbarium collections and studying the Amaranthaceae family.

Engage the reader by highlighting intriguing findings or potential discoveries that may emerge from the analysis of this extensive herbarium collection.

Some references are incomplete.

Consider concluding the abstract with a statement that underscores the broader implications of the research and its potential contributions to scientific knowledge and practical applications.

By addressing these points, the paper can be enhanced to provide a clearer, more informative overview of the research conducted on the herbarium collection of Amaranthaceae specimens. Recommended for publication after minor revision.

Comments on the Quality of English Language

Moderate English revision is required 

Reviewer 3 Report

Comments and Suggestions for Authors

Dear Prof. Dr. Sunny Diao

The manuscript entitled "Geographical-historical analysis of the herbarium specimens representing the economically important family Amaranthaceae
(Chenopodiaceae-Amaranthaceae clade) collected in years 1821-2022 and
preserved in the Herbarium of the Jagiellonian University in Kra" has been reviewed. The subject is interesting and helpful for readers. A few comments have been cited in the text. 

Sincerely Yours 

Reviewer 4 Report

Comments and Suggestions for Authors

Here's the updated review report with the explanation that the manuscript can only be considered as a short communication paper:

Review Report:

The manuscript titled "Geographical-historical analysis of the herbarium specimens representing the economically important family Amaranthaceae (Chenopodiaceae-Amaranthaceae clade) collected in years 1821-2022 and preserved in the Herbarium of the Jagiellonian University in Kra" presents an analysis of the herbarium collection of the Amaranthaceae family at the Jagiellonian University Herbarium. However, in its current form, the study does not meet the standards for publication in a high-ranking (Q1) journal in the field of biology.

The main concerns are:

1. Lack of novelty and significance: The study primarily documents and analyzes existing herbarium specimens without presenting novel findings or advancements in knowledge that would be expected in a top-tier journal.

2. Limited scope and depth: The manuscript focuses solely on the collection details and origins of the specimens, without delving into more substantive aspects such as taxonomic revisions, phylogenetic analyses, or comprehensive studies on the economic importance or utilization of the Amaranthaceae family.

3. Absence of rigorous methodology: The abstract does not provide details on the methodological approaches employed, such as data analysis techniques, statistical methods, or experimental designs, which are crucial for evaluating the study's robustness.

4. Lack of specific research questions or hypotheses: The abstract does not clearly articulate specific research questions or hypotheses that the study aims to address, which is typically expected in research articles published in top-tier journals.

5. Preliminary nature: The abstract reads more like a descriptive report or preliminary study, rather than a complete and comprehensive research article suitable for publication in a Q1 journal.

To be considered for publication in a high-ranking journal, the authors would need to significantly expand the scope and depth of the study, incorporate more rigorous methodologies, address specific research questions or hypotheses, and highlight the novel contributions or significant implications of their findings within the field of study.

In its current form, I recommend that the manuscript be rejected for publication as a full research article in a Q1 journal in the field of biology. However, the manuscript could potentially be considered for publication as a short communication paper, provided that the content is focused and concise, and the authors clearly outline the specific contribution or purpose of documenting and analyzing this herbarium collection.

Comments on the Quality of English Language

Extensive editing of English language is required
